# RiTTA: Modeling Event Relations in Text-to-Audio Generation

## Abstract

Despite significant advancements in Text-to-Audio (TTA) generation models achieving high-fidelity audio with fine-grained context understanding, they struggle to model the relations between audio events described in the input text. However, previous TTA methods have not systematically explored audio event relation modeling, nor have they proposed frameworks to enhance this capability. In this work, we systematically study audio event relation modeling in TTA generation models. We first establish a benchmark for this task by: (1) proposing a comprehensive relation corpus covering all potential relations in real-world scenarios; (2) introducing a new audio event corpus encompassing commonly heard audios; and (3) proposing new evaluation metrics to assess audio event relation modeling from various perspectives. Furthermore, we propose a finetuning framework to enhance existing TTA models' ability to model audio events relation.

## 1 Introduction

Text-based crossmodal content generation has gained significant attention in recent years as it opens up new possibilities for even amateur users to create professional content. Typical such methods include text-to-image (TTI) (Ho et al., 2020), text-to-music (TTM) (Copet et al., 2023), text-to-point (TTP) (Nichol et al., 2022), text-to-speech (TTS) (Ren et al., 2019) text-to-audio (TTA) (Liu et al., 2024; Huang et al., 2023b). Among all of them, text-to-audio (TTA) generation stands out as a particularly promising area, enabling the synthesis of complex acoustic environments or soundscapes directly from textual descriptions. Recent advances in TTA have demonstrated impressive progress in generating high-quality, detail-rich audio described in the input text prompt (Liu et al., 2024; 2023a; Huang et al., 2023b;a; Ghosal et al., 2023; Majumder et al., 2024; Kreuk et al., 2023).

When perceiving the physical world acoustically, whether through text or audio, the fundamental unit is the audio event, a distinct acoustic signal representing an independent source. The essence of perception lies in understanding the relationships emerging from events. Audio events are spatiotemporally distributed in the physical world. Together with relation, they contribute for holistic acoustic scene understanding (Qu et al., 2022). Studies in psychology (Zacks et al., 2007) and neuroscience (Lake et al., 2015; Hirsh et al., 1967) show that the human brain perceives the environment through discrete events and the relations between them. Humans are adept at using rich language to describe both audio events and their intricate relationships. While current TTA models can generate audios with high fidelity, their ability to generate audios that not only includes audio events but also preserves the text-informed relationships between them remains unexplored.

As a primary study, we prompt the latest six TTA models with an exemplar text with explicit audio events and their relation *generate dog barking audio, followed by cat meowing audio*. Next we check if the specified audio events are present and if so, their relations are correct in the generated audios. As is shown in Table 1, all existing TTA models fail to properly model temporal relationships in the generated audio, even when they succeed in generating the correct audio events. The generated audio waveform, spectrum and another case study with a

| Text Prompt: generate dog barking audio, followed by cat meowing audio | | |
|---|---|---|
| Method | Relation? | Remark |
| AudioLDM (2023a) | ✗ | just cat meow, low-fidelity |
| AudioLDM 2 (2024) | ✗ | output dog barking |
| MakeAnAudio (2023b) | ✗ | just cat meow, low-fidelity |
| AudioGen (2023) | ✗ | output wrong audios |
| Tango (2024) | ✗ | two audios, low fidelity |
| Tango 2 (2024) | ✗ | can output two audios |

Table 1: A case study on relation of TTA methods. Listenable audios are provided in suppplementary material.

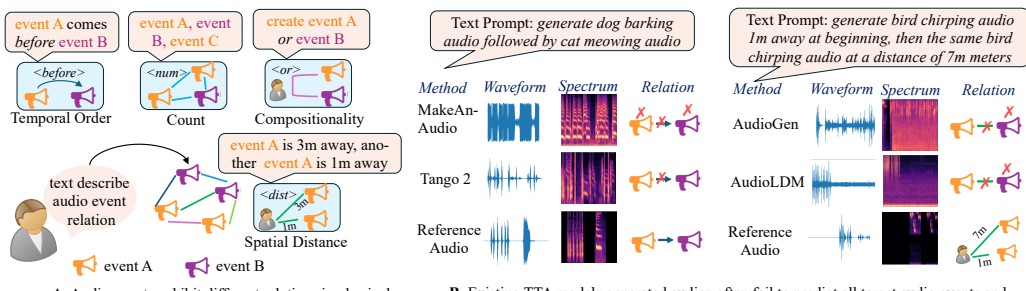

Figure 1: *RiTTA* Motivation: The acoustic world is rich with diverse audio events that exhibit various relationships. While text can precisely describe these relationships (Fig. A), current TTA models struggle to capture both the audio events and the relations conveyed by the text (Fig. B). This challenge motivates us to systematically study *RiTTA*.

much complex text are shown in Fig. 1. The poor performance of current TTA models in modeling audio events relation, along with the lack of systematic discussion on this topic, motivates us to explore *Relation in TTA* (dubbed *RiTTA*) in depth in this work. We visualize the motivation in Fig. 1.

To systematically study *RiTTA*, we first benchmark it from four key perspectives: 1. we construct a comprehensive audio event relation corpus that captures common relationships found in the physical world. Unlike visual relations in cross-modal image tasks, which mainly focus on spatial aspects (*e.g.*, left, bottom) (Gokhale et al., 2022), audio events exhibit far more complex relationships spanning spatial, temporal, and compositional dimensions. Consequently, we define four primary relation categories: *Temporal Order*, *Spatial Distance*, *Count*, and *Compositionality*. 2. Accompanying the relation corpus, we build an audio event category corpus derived from five main sources, each of which is further linked to multiple seed audios. 3. devise a `<textprompt,audio>` pair generation strategy emphasising both text prompt and audio diversity. 4. propose a new relation aware evaluation framework that assesses the relation in a multi-stage manner. The proposed benchmark will benefit the community to explore *RiTTA* in greater depth. Additionally, we introduce a fine-tuning strategy based on the latest state-of-the-art (SoTA) TTA model and demonstrate its effectivenss in relation modelling. In summary, we make the following four main contributions:

1. We conduct an extensive evaluation of existing TTA models in modeling the audio events relations and demonstrate their inability to capture these relations in the generated audios.

2. We benchmark *RiTTA* by constructing complete relation corpus, audio event category corpus, seed audio corpus. Combined with the `<textprompt,audio>` pair generation strategy, researchers can create large, diverse dataset to further investigate the *RiTTA* task.

3. We propose a new multi-stage relation aware evaluation framework, called *MSR-RiTTA*, which offers a more nuanced evaluation compared to existing TTA metrics, allowing researchers to quantitatively assess their methods from multiple angles.

4. We introduce a fine-tuning strategy leveraging the new dataset, demonstrating improvement over current SoTA methods.

## 2   RELATED WORK

**Audio Generation** has received lots of attention and made significant progress in recent years, advanced by fast-progressing generative AI technologies (Ho et al., 2020; Rombach et al., 2022). Audio generation encompasses sub-tasks such as text-to-speech (TTS) that focuses on generating speech from text transcription (*e.g.*, FastSpeech (Ren et al., 2019) and GradTTS (Popov et al., 2021)), text-to-music (TTM) that generates music from text input (*e.g.*, MusicLM (Agostinelli et al., 2023), MusicGen (Copet et al., 2023)) and Image-to-Audio (ITA) generation that generates audio from image input (*e.g.*, Img2Wav (Sheffer & Adi, 2023), SpecVQGAN (Iashin & Rahtu, 2021), RegNet (Chen et al., 2020)) and Text-to-Audio (TTA) generation aiming to generate corresponding audio described by text (*e.g.*, AudioLDM (Liu et al., 2024; 2023a; Yang et al., 2022), DiffSound (Yang et al., 2022)).

**Text-to-Audio (TTA) Generation** involves producing audio that faithfully reflects the acoustic content or behavior described by the input text. Recent advancements have significantly improved the

| Main Relation | Sub-Relation | Sample Text Prompt |
|---|---|---|
| Temporal Order | before; after; simultaneity | generate dog barking audio, followed by cat meowing; |
| Spatial Distance | close first; far first; equal dist. | generate dog barking audio that is 1 meter away, followed by another 5 meters away. |
| Count | count | produce 3 audios: dog barking, cat meowing and talking. |
| Compositionality | and; or; not; if-then-else | create dog barking audio or cat meowing audio. |

| Main Category | Sub-Category |
|---|---|
| Human Audio | baby crying; talking; laughing; coughing; whistling |
| Animal Audio | cat meowing; bird chirping; dog barking; rooster crowing; sheep bleating |
| Machinery | boat horn; car horn; door bell; paper shredder; telephone ring |
| Human-Object Interaction | vegetable chopping; door slam; footstep; keyboard typing; toilet flush |
| Object-Object Interaction | emergent brake; glass drop; hammer nailing; key jingling; wood sawing |

Table 2: Audio Events Relation Corpus.

Table 3: Audio Events Category Corpus.

quality and intelligibility of generated audio (Liu et al., 2024; 2023a; Kreuk et al., 2023; Yang et al., 2022; Ghosal et al., 2023; Liao et al., 2024). Despite improvements in audio quality and intelligibility, existing TTA methods still lag significantly in their ability to model relationships between audio events in the generated audio. AudioLDM (Liu et al., 2023a) builds on latent space (Rombach et al., 2022) to learn continuous representation.

**Audio Events Relation Modelling**. In the context of environmental audio, a set of audio events exhibit relationships that are crucial for holistic acoustic scene understanding. Based on how audio interact with the physical world in space, time and perceptual aspects, the resulting audio events exhibit complex relationships in spatial, temporal and compositional aspects. Prior work has partially addressed modeling certain temporal relations (*e.g.*, order) in TTA (Xie et al., 2024) and compositional reasoning (Ghosh et al., 2024) for discriminative tasks, such as audio classification and audio-text retrieval. WavJourney (Liu et al., 2023b) leverages a large language model alongside multiple audio generation models to achieve compositional audio generation. However, its limitations include an artificial post-mixing process, which may result in generated audio lacking smooth transitions across event boundaries and inefficiencies in inference. While prior research has touched on modeling audio event relations, their potential in TTA remains largely underexplored. If we analogize an audio event to an object in image, the corresponding relationships exhibited in an image are mainly limited to 2D spatial relationship (*e.g.*, before, bottom, left). Despite object of interest spatial relationship learning and evaluation have received lots of attention in recent years (Krishna et al., 2016; Gokhale et al., 2022; Okawa et al., 2023), the research on audio event relation modelling has been almost ignored.

## 3 BENCHMARK TTA AUDIO EVENTS RELATION

In this section, we sequentially present audio events relation corpus in Sec. 3.1, audio event category corpus in Sec. 3.2, seed audio corpus and `<textprompt,audio>` pair generation strategy in Sec. 3.3. Finally, the relation aware evaluation framework *MSR-RiTTA* is presented in Sec. 3.4.

### 3.1 AUDIO EVENT RELATION CORPUS

An audio event refers to a distinct acoustic signal occurrence with specific frequency, duration and context characteristics that can be attributed to distinguish an independent sound source (He et al., 2021) in an environment. Audio event is ubiquitous in the physical world and serves as the fundamental entity to analyze and interpret the acoustic scene. We embrace the audio event as the fundamental element to construct the relation corpus.

We construct the audio events relation corpus based on two key aspects. First, we consider relations commonly found in the physical world, such as those arising from spatial and temporal variations, which test TTA models' ability to replicate audio events' interactions in real-world scenarios. Second, we focus on relations that challenge TTA models' logical reasoning, evaluating their ability to determine both which audio events to generate and how to generate them. These two aspects partially overlap. Specifically, we define five main audio event categories, each associated with five subcategories of audio events. The detailed relation corpus is provided in Table 2, including,

1. **Number Count**: The number of audio events included in the generated audio, testing TTA models' ability to address acoustic polyphony challenge.

2. **Temporal Order**: Temporal order refers to the sequence of audio events in the generated audio. We include three basic temporal relations for two audio events: `before`, `after`, and `simultaneity`, testing the TTA models' ability to distinguish and generate the correct event order as specified in the input text prompt.

3. **Spatial Distance**: Spatial distance refers to the variation in relative spatial distances inferred from the generated audio. It evaluates the TTA models' ability to capture the spatial distance differences specified in the text prompt. Since we focus on mono-channel audio, obtaining the absolute distance for each audio event is nearly impossible (He et al., 2021). Therefore, we rely on loudness differences within intra-class audio events to verify their spatial distance variations.

4. **Compositionality**: Compositionality relation describes how multiple individual audio events are integrated together to form a complex auditory structure that specified in the input text prompt. It tests TTA models' logical reasoning capability in determining which audio events to generate and how to structure them, by following the compositional guidance illustrated in the input text prompt. Specifically, we incorporate four main compositionality relations: Conjunction (`And`, *e.g.*, generate audio A and audio B together); Disjunction (`Or`, *e.g.*, generate audio A or Audio B, not both); Negation (`Not`, exclude one particular audio event, *e.g.*, do not generate dog barking audio); Condition (`if-then-else`, either generate two audio events if the condition is met, otherwise generate the third audio if the condition is not met).

Most of the relations relate to two audio events (see Table 3 for more detail). Expanding the corpus to include more complex relations with a greater number of audio events is left for future work.

## 3.2 AUDIO EVENT CATEGORY CORPUS

Alongside the relation corpus presented in Sec. 3.1, we further construct a comprehensive audio event category corpus. The two corpora serve as fundamental dataset for constructing text prompts for TTA models. Since different audio event signals are generated from various sources or through different interactions, we first establish four main audio source categories, further detailing each category with five sub-categories. These constructed audio categories encompass the majority of ubiquitous audio events encountered in our daily lives. Specifically, the audio event category corpus contain,

1. **Human Audio**: the audio generated by human beings in our daily life, including *baby crying*, *coughing*, *laughing*, *whistling*, *female speech* and *male speech*.

2. **Animal Audio**: the audio generated by animals, including *cat meowing*, *dog barking*, *bird chirping*, *horse neighing*, *rooster crowing*, *sheep bleating* and *pig oinking*.

3. **Machinery Audio**: the audio generated by various machinery devices while they are working, including *car horn*, *doorbell*, *telephone ring*, *paper shredder* and *boat horn*.

4. **Human-Object Interaction Audio**: human-object interaction audios include *vegetable chopping*, *keyboard typing*, *toilet flushing*, *door slamming* and *foot step*.

5. **Object-Object Interaction Audio**: we further incorporate object-object interaction audios, including *glass dropping*, *car emergency brake*, *hammering nail*, *wood sawing* and *keys jingling*.

The detailed audio event corpus is given in Table 3. With the constructed relation and audio event corpus, we can create relation aware text prompts for TTA models.

## 3.3 SEED AUDIO CORPUS AND TEXT-AUDIO PAIR CREATION STRATEGY

In order to create the corresponding audio for any constructed text prompt, we instantiate each audio event presented in Sec. 3.2 with five exemplar seed audios collected from `freesound.org` [1]. Since most audio files on `freesound.org` are uploaded by volunteers who recorded them in their daily lives, incorporating five exemplar audios for each individual audio event category enhances both the diversity and realism of the seed audio. For instance, in the case of the `dog barking` audio event, the five selected audios vary in terms of dog breeds and barking styles. To further enhance an

---

[1] since `freesound.org` does not contain meaningful people talking audio, we collect people talking audio from VCTK (Yamagishi et al., 2019)

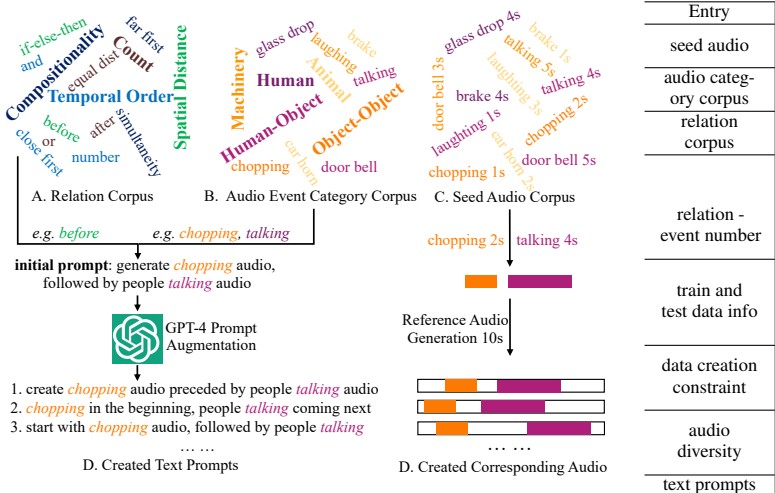

Figure 3: Relation aware `<textprompt,audio>` pair creation pipeline. It introduces large diversity in both text prompt and audio.

| Entry | Highlight |
|---|---|
| seed audio | one event has 5 audios each has 1 s-5 s audio clips |
| audio category corpus | 5 main categories 25 sub-categories |
| relation corpus | 4 main 11 sub relations |
| relation - event number | `count`: 2-5 events; `Not`: 1 event; `if-then-else`: 3 events others: 2 events. |
| train and test data info | each audio is 10 s long sampling rate 16 kHz train: 44 hrs, 1.6 k pairs test: 22 hrs, 0.8 k pairs |
| data creation constraint | `count` inter-category *SpatialDist* intra-category and require temporder |
| audio diversity | one event → multi-audios; seed audio → multi time len; seed audios various start time |
| text prompts diversity | GPT-4 augmented prompts; one template → multi-events. |

Table 4: *RiTTA* benchmark highlights.

audio event's temporal length diversity, we randomly slice each seed audio into non-overlapping clips ranging from 1 sec to 5 secs. In summary, we have constructed 11 relations (see Table 2 Sub-Relation column), and 25 audio events across five main audio events categories. Each audio event has been associated with 5 diverse audio clips ranging from 1 sec to 5 secs collected from `freesound.org`.

**Text Prompt Generation**: a proper audio events relation aware text prompt comprises of two parts: a relation (*e.g.*, `<before>`) and audio events categories. The audio event categories can be either intra-class or inter-class, and the audio event number depends on the relation. We first instantiate an initial text prompt describing this relation. For example, for the temporal order `before` relation, the

```
1. generate audio A succeeded by B;
2. start with A, followed by B;
3. play A initially, B afterwards;
4. generate A preceded by B;
5. A in the beginning, B coming next;
```

Figure 2: GPT-4 augmented prompts (`before` relation).

initial text prompt can be like: *generate audio A, followed by audio B*. To enrich the text prompts, we further use the initial text prompt to query LLM (in our case GPT-4) to provide more text prompts with diverse descriptive language for the same relation. One such GPT-4 augmented text prompts is shown in Fig. 2, which illustrates that the same relation can be exactly expressed by multiple different text prompts. By incorporating GPT-4, we create 5 text prompts for each individual relation.

**Audio Generation**: Given the aforementioned audio events categories and relation, we randomly select an exemplar seed audio for each audio event and further linearly blend them together by satisfying the specified relation. For example, the relation `<before>` requires two audio events, the two selected audios can be blended together to form the final audio as long as the two seed audios satisfy the `<before>` relation (Fig. 3, D). Notably, unlike blending two objects in an image that requires careful consideration of factors like occlusion and viewing angle, combining two audio signals simply involves linearly adding them together (Pierce, 2019). This offers an advantage for audio generation, as it eliminates the need for additional operations beyond the specified relation.

The generation of the `<textprompt,audio>` pair is further illustrated in Fig. 3. With the proposed `<textprompt,audio>` pair generation strategy, we can create massive diverse pairs even for the same audio events and the same relation, significantly enhancing the diversity and generalization capability of our generated dataset.

### 3.4 RELATION AWARE EVALUATION METRIC MSR-RITTA

Existing TTA methods adopt general evaluation metrics to asses the similarity between generated audio and reference audio, including Fréchet Audio Distance (FAD), Fréchet Distance (FD) (Heusel et al., 2017), Kullback–Leibler (KL) divergence, Fréchet Inception Distance (FID) *etc.*, among others. While those general evaluation metrics give an overall estimation of the similarity between the two

comparing audios, they do not offer direct relation-aware evaluations. In addition to incorporating general evaluation metrics, we further propose multi-stage relation-aware evaluation metrics, with which we can gain insight on how the method performs w.r.t. difference relations.

**General Evaluation Metric**: We incorporate three widely used general evaluation metrics: the objective evaluation metric FAD, FD and KL divergence scores. FAD and FD measure the distribution similarity with feature embedding extracted from pre-trained on VGGish model (Hershey et al., 2017).

**Relation aware Evaluation Metric MSR-RiTTA**: To directly measure how accurately the text-indicated relation is reflected in the generated audio, we incorporate relation aware metrics for each specified relation. In relation aware evaluation, we base on the individual audio event to compute the metrics, which allows us to measure the relation between audio events. Let's denote $(\mathcal{A}_g, \mathcal{T}, \mathcal{R}, \mathcal{A}_p)$ by ground truth audios, text prompts, relations and generated audios, respectively. We first extract audio events $\mathcal{E}$ from generated audios $\mathcal{A}_p$. For example, for the $i$-th generated audio $a_i^p$, we apply pre-trained audio event detection model (we use finetuned PANNS (Kong et al., 2020b), see Sec. .1 in Appendix) to extract all potential audio events involved in the audio $E_{a_i^p} = \{(e_j, m_j)|s\}_{i=1}^k$ by a given event confidence threshold $s \in \mathcal{S}$, where $e_j$ is the $j$-th audio event and $m_j$ is the corresponding meta data (*e.g.*, audio event class label, confidence score, temporal start time and end time, see Fig. 4). To obtain audio events data for ground truth audios, we can either apply the same pre-trained model or directly extract from text prompts. Finally, we can get $(\mathcal{A}_g, \mathcal{T}, \mathcal{R}, \mathcal{A}_p, \mathcal{E}_p, \mathcal{E}_g)$, the relation aware evaluation function $f(\cdot)$ depends on the audio events $\mathcal{E}_p$, $\mathcal{E}_g$ and relations $\mathcal{R}$, $f(\mathcal{E}_p, \mathcal{E}_g|\mathcal{R}, s)$. We adopt a multi-stage relation (*MSR-RiTTA*) aware evaluation strategy.

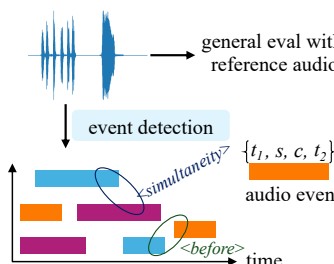

Figure 4: relation aware evaluation. Audio event detection model is applied to get audio events. The meta data of each event contains start time $t_1$, end time $t_2$, confidence score $s$ and class label $c$. Various relations can be discovered from these audio events.

**Stage 1:** Target Audio Events Presence (**Pre**). The paramount requirement for a successful audio generation is the presence of text-specified audio events in the generated audio. In this evaluation, the ground truth audio events and generated audio events are treated as *set*. For a given ground truth and generated audio events pair $(E_g, E_p)$, we iterate over each audio event $e_g$ in the ground truth $E_g$ to check if it exists in the generated audio events $E_p$, regardless of its number and temporal position.

$$f_p(E_p, E_g) = \frac{1}{k} \sum_{e_g \in E_g} \mathbb{1}(e_g, E_p); \quad \mathbb{1}(e_g, E_p) = \begin{cases} 1, \text{if } e_g \in E_p \\ 0, \quad \text{otherwise}, \end{cases} \tag{1}$$

where $k$ is audio event number in the ground truth. $s_l(e_g)$ is a potential event meeting the confidence threshold in the generated audio. We select the event with the highest confidence score as the target.

**Stage 2:** Relation Correctness (**Rel**). Once confirming the aforementioned target audio presence, we further investigate if these audio events obey text-specified relation. The relation is correctly modelled if at least a subset of generated audio events meet the relation. We give score 1 if relation is correctly modelled, otherwise score 0.

$$f_r(E_p|R) = \prod_{E_t \in E_p \cap E_g} \mathbb{1}(E_t, R); \quad \mathbb{1}(E_t, R) = \begin{cases} 1, \quad \text{if } E_t \text{ satisfies relation } R, \\ 0, \quad \text{otherwise}, \end{cases} \tag{2}$$

**Stage 3:** Audio Parsimony (**Par**). Apart from requiring to generate all target audios, we should discourage the model from generating excessive intra-class audio events or irrelevant inter-class audio events. We call this property *Audio Parsimony*. Once it is violated, we introduce extra penalty.

$$f_s(E_p, E_g) = \exp\left(-w_s \cdot |n(E_p) - n(E_g)|\right) \tag{3}$$

where $n(\cdot)$ indicates audio event number. $w_s$ is the weight adjusting the penalty (in our case, $w_s = 0.1$). The higher audio event number difference incurs lower parsimony score, the resulting parsimony score lies within $(0, 1)$. The final relation aware score based on the audio event confidence threshold $s$ equals to the multiplication of the three stage scores,

$$f(\mathcal{E}_p, \mathcal{E}_g|\mathcal{R}, s) = \frac{1}{N} \sum_{(E_p, E_g, R) \in (\mathcal{E}_p, \mathcal{E}_g, \mathcal{R})} f_p(E_p, E_g) \cdot f_r(E_p|R) \cdot f_s(E_p, E_g) \tag{4}$$

where $N$ is data size number. The final average MSR (AMSR) score $f(\mathcal{E}_p, \mathcal{E}_g | \mathcal{R}, s)$ lies within $[0, 1)$ (the higher of the score, the better of the model's performance). Following prior COCO object detection evaluation strategy (Lin et al., 2014), we further average across multiple discrete audio event confidence thresholds to get the mean average MSR score (mAMSR), $f(\mathcal{E}_p, \mathcal{E}_g | \mathcal{R})$,

$$f(\mathcal{E}_p, \mathcal{E}_g | \mathcal{R}) = \frac{1}{K} \sum_{s \in \mathcal{S}} f(\mathcal{E}_p, \mathcal{E}_g | \mathcal{R}) \tag{5}$$

where $K$ is the discrete audio event confidence thresholds number. In our case we use uniformly sample four confidence thresholds in range $[0.5, 0.8]$ with step size $0.1$.

## 4    RELATION AWARE TTA FINETUNING

Existing TTA models adopt audio-language pre-trained model to extract text and audio embeddings, including CLAP (Wu et al., 2023b) and FLAN-T5 (Chung et al., 2024). Prior work (Ma et al., 2023; Yuksekgonul et al., 2023; Wu et al., 2023a; Ghosh et al., 2024) show that existing audio-language pre-trained models (*e.g.*, CLAP (Wu et al., 2023b)) performs like bag-of-words (BoW), which means they are far better at audio event retrieval task than audio events temporal relation task. Moreover, the dataset used to pre-train audio-language such as AudioSet (Gemmeke et al., 2017) and AudioCaps (Kim et al., 2019) are dominated by unary audio event ($64\%$ (Ghosh

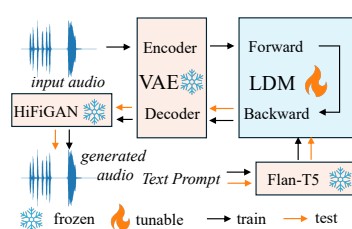

Figure 5: RiTTA finetune pipeline.

et al., 2024)), limiting models from learning meaningful representations for audio event relations.

Based on aforementioned discussion, we propose to finetune the existing latest Tango model (Ghosal et al., 2023) with our created relation aware dataset (we finetuned Tango 2 as well, but found it gave inferior performance than Tango). Tango depends on prior TTA frameworks AudioLDM (Liu et al., 2023a) to use a Variational Autoencoder (VAE) for audio encoding and decoding, a latent diffusion model (LDM) (Rombach et al., 2022) for audio generation and HiFiGAN (Kong et al., 2020a) to generate final audio waveform from VAE decoder decoded mel-spectrogram. Unlike AudioLDM (Liu et al., 2023a) which depend on CLAP (Wu et al., 2023b) for text prompt encoding, Tango adopts pre-trained Flan-T5 (Chung et al., 2024) model for text prompt encoding. Latest TTA models such as Tango (Ghosal et al., 2023), Tango 2 (Majumder et al., 2024) and AudioLDM 2 (Liu et al., 2024) show that Flan-T5 can achieve better performance than CLAP (Wu et al., 2023b) in TTA task. Benefiting from the latest advancement, we fine-tune Tango by just tuning latent diffusion model (LDM) and fixing VAE, HiFiGAN and Flan-T5 components. In our case, we finetune Tango with the curated 44 hrs training dataset. The finetuning workflow is shown in Fig. 5 and finetuing detail in Sec. 4.

## 5    EXPERIMENT

We run two experiments: benchmarking existing TTA methods on our curated 22 hrs benchmark dataset (aka testing dataset). Fine-tuning the advanced TTA model on our curated 44 hrs training dataset and further test its relation modelling capability.

### 5.1    MORE DISCUSSION ON DATA CREATION

We follow the strategy presented in Sec. 3.3 to create the dataset. Specifically, for each of the 11 sub-relations in Table 2, we create 720 (2 hrs audio) `<textpromt,audio>` pairs for testing (aka benchmark dataset) and 1440 pairs (4 hrs audio) for training (aka finetuning dataset). The highlight of the training/testing dataset is given in Table 4.

To ensure that all relations can be effectively evaluated using our method, we applied two key constraints during the data creation process. First, to make the audio events countable without ambiguity, we selected inter-category audio events to form the `<textprompt,audio>` pairs. This avoids the ambiguity that arises when using intra-category events, especially for those with repetitive, similar local occurrences (*e.g.*, multiple instances of dog barking). Second, for the *Spatial Distance* relation, we introduced a temporal order constraint to ensure that the two audio events do not overlap in time. Temporal overlap would require complex source separation models (Petermann et al., 2023) to distinguish individual events. By enforcing this non-overlapping constraint, the evaluation of *Spatial Distance* becomes manageable using an audio event detection model (see Sec. A in Appendix). The basic information of data creation is given in Table 4.

Table 5: Benchmark quantitative result across all relations. mAPre, mARel and mAPar are in $10^{-2}$. mAPre and mARel can be treated as *presence*, *relation correctness* percentage ratio, they lie in range $[0, 100]$. mAPar score also lies within $[0, 100]$. mAMSR ($10^{-4}$) lies in range $[0, 1]$. The top-, second- and third- performing methods are labelled in different colors, respectively.

| Model | #param | General Evaluation | | | Relation Aware Evaluation (↑) | | | |
|---|---|---|---|---|---|---|---|---|
| | | FAD ↓ | KL ↓ | FD ↓ | mAPre | mARel | mAPar | mAMSR |
| AudioLDM (S-Full) (2023a) | 185 M | 5.65 | 38.95 | 37.30 | 2.76 | 0.50 | 2.52 | 0.04 |
| AudioLDM (L-Full) (2023a) | 739 M | 5.47 | 38.42 | 37.96 | 3.09 | 0.77 | 2.56 | 0.08 |
| AudioLDM 2 (L-Full) (2024) | 844 M | 6.68 | 29.07 | 35.85 | 12.26 | 2.41 | 10.01 | 3.39 |
| MakeAnAudio (2023b) | 452 M | 9.46 | 82.72 | 45.98 | 8.14 | 1.68 | 6.47 | 1.02 |
| AudioGen (2023) | 1.5 B | 6.43 | 28.01 | 32.04 | 9.61 | 2.12 | 8.60 | 2.27 |
| Tango (2023) | 866 M | 10.79 | 90.26 | 39.46 | 11.13 | 2.27 | 9.88 | 3.10 |
| Tango 2 (2024) | 866 M | 13.84 | 89.66 | 44.03 | 16.63 | 4.40 | 12.53 | 11.55 |

Table 6: Benchmark quantitative result w.r.t. the four main relations. We report FAD sore and mAMSR score for general evaluation and relation aware evaluation, respectively.

| Model | General Evaluation (FAD ↓) | | | | Relation Aware Eval. (mAMSR ↑) | | | |
|---|---|---|---|---|---|---|---|---|
| | Count | TempOrder | SpatDist | Compos | Count | TempOrder | SpatDist | Compos |
| AudioLDM (S-Full) (2023a) | 3.85 | 6.86 | 4.56 | 9.36 | 0.00 | 0.05 | 0.00 | 0.18 |
| AudioLDM (L-Full) (2023a) | 3.68 | 6.45 | 4.10 | 8.98 | 0.00 | 0.05 | 0.06 | 0.17 |
| AudioLDM 2 (L-Full) (2023b) | 5.03 | 8.94 | 4.72 | 9.41 | 0.14 | 1.87 | 1.46 | 9.89 |
| MakeAnAudio (2023b) | 6.02 | 10.21 | 8.18 | 12.78 | 0.12 | 0.66 | 0.44 | 2.40 |
| AudioGen (2023) | 6.14 | 8.39 | 3.38 | 9.98 | 0.32 | 3.83 | 0.48 | 4.18 |
| Tango (2023) | 8.54 | 10.25 | 10.11 | 13.97 | 0.16 | 3.44 | 0.82 | 8.10 |
| Tango 2 (2024) | 10.01 | 13.91 | 13.23 | 17.04 | 0.96 | 20.92 | 1.92 | 23.25 |

## 5.2 MORE DISCUSSION ON RiTAA EVALUATION

Section 3.4 has introduced the metrics in general. In practice, we further adjust the audio generation process for relations under *Compositionality* and *Spatial Distance* to so as to ensure these relations can be accurately evaluated under our proposed framework.

First, we skip general evaluation for `<Not>` as it lacks a corresponding ground truth reference audio. During fintuning, we generate silent audio for `<Not>` for create finetuing pairs. Second, for the `<if-then-else>` and `<Or>` sub-relations, which correspond to two possible ground truth audios, we handle evaluation by computing the L2 distance (in the time domain) between the generated audio and the two reference audios. For example, for the prompt *if event A then event B, else event C*, the first reference is the combination of events A and B, while the second contains only event C. We use the reference audio with smaller L2 distance to the generated audio for general evaluation.

Third, precise evaluation of the three sub-relations (`<closefirst>`, `<farfirst>`, and `<equaldist>`) under *Spatial Distance* from unconstrained audio requires sound event detection and localization (SELD (He & Markham, 2023; Grondin et al., 2019)) techniques to spatially localize each audio event, which is impossible with mono-channel audio. To address this, we approximate spatial distance by calculating the loudness, which can be estimated using the L2 norm of the audio waveform. The rationale behind this approach is that greater distances result in a dampening of waveform amplitude (and consequently reduced loudness) due to energy decay along the audio propagation path. When the loudness difference exceeds a predefined threshold (for `<closefirst>`, `<farfirst>`) or is within that threshold (for `<equaldist>`), we consider the evaluation accurate. Specifically, we use a loudness reduction ratio $\sigma_1$ (with $\sigma_1 = 0.2$ in our case). For `<closefirst>`, if the closer event's loudness is at least $\sigma$ times greater than the further event's loudness, the relation is considered correct. Similarly, for `<equaldist>`, the loudness difference between the two events should be within $\sigma_2$ (with $\sigma_2 = 0.4$ in our case) of the louder event's loudness. This estimation is also reflected in the data generation process (see Sec 5.1).

## 5.3 RELATION AWARE BENCHMARKING RESULT

We benchmark our curated test dataset on 7 most recent TTA models: AudioLDM (Liu et al., 2023a) (two versions), AudioLDM 2 (Liu et al., 2024), MakeAnAudio (Huang et al., 2023b),

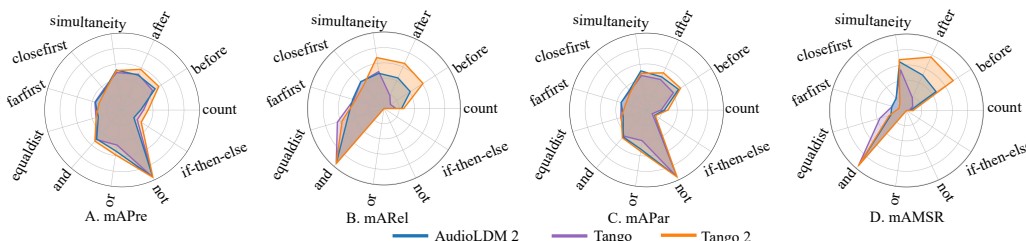

Figure 6: Top 3 performing in audio events relation modelling TTA methods' performance w.r.t. the 11 sub-relations. We report mAPre, mARel, mAPar and mAMSR scores separately.

AudioGen (Kreuk et al., 2023), Tango (Ghosal et al., 2023) and Tango 2 (Majumder et al., 2024). We directly depend on their released models to generate a 10 second audio from each text prompt. We then adopt general evaluation and relation-aware evaluation metrics (see Sec. 3.4) for assessing the generated audios quality. The detailed configuration of each method is given in Table I in Appendix.

The quantitative evaluation results across all relations are shown in Table 5. From this table we can observe that the general evaluation results are inconsistent with our proposed relation aware evaluation metrics. The best performing methods under generational evaluations (the two AudioLDM versions) perform the worst under relation aware evaluations, and vice versa. These discrepancies highlight the necessity of proposing evaluation metrics specifically tailored for audio events relations. Additionally, while the performance differences among the seven benchmarking methods under general evaluation are relatively minor, the corresponding differences under relation aware evaluation are significantly more pronounced (*e.g.*, Tango 2 outperforms AudioLDM (S-Full) by about 200 times). However, even the top-performing method, Tango 2 (Majumder et al., 2024), still struggles to model audio events relations, as both its presence accuracy and relation accuracy rate are below 1% (mAPre is just 0.02% and mARel 0.04%), and it generates an average of two redundant audio events (mAPar=0.1253). All of these observations demonstrate the limitations of existing TTA methods in modelling audio events relation and the necessity to systematically study audio events relation in TTA, highlighting the importance of our proposed work.

The quantitative evaluation results (mAMSR score) w.r.t the four main relation categories are presented in Table 6. We observe that both general and relation-aware evaluations show better performance on *Temporal Order* and *Compositionality* compared to *Count* and *Spatial Distance*. This suggests that the *Count* and *Spatial Distance* relations pose significant challenges for TTA tasks. Additionally, we visualize the detailed relation aware evaluation results for the 11 sub-relations, highlighting the top three performing methods AudioLDM 2 (Liu et al., 2024), Tango (Ghosal et al., 2023), and Tango 2 (Majumder et al., 2024), in Fig. 6. We can observe that all the three methods 1. achieve exceedingly high presence score on `Not` relation, which is expected since a high **Presence** score (Subfig. A) can be easily obtained by simply not generating the specified audio event. 2. perform well in modelling `And` relation (Subfig. B) (then `<equaldist>` and the three relations in *Temporal Order*); 3. exhibit strength in generating concise audios particularly for `Not` relation (Subfig. C). Overall, all the three methods excel in modelling `And` relation and then the three sub-relations in *Temporal Order*, which is also reflected by the result in Table 6.

The key findings from the relation-aware benchmarking are summarized in the Table 7. In summary, we conclude that, 1. existing TTA models lack the ability to model audio events relation described by the text prompt in the generated audio, emphasizing the importance of our work in systematically study audio events relation in TTA. 2. Existing TTA evaluation metrics fall short in accurately measuring audio events relations from the generated audio. Our proposed multi-stage relation evaluation framework suffices to measure the relation accuracy from various aspects.

**1.** generation eval. contradicts with RiTTA eval.
**2.** *TemOrder/Compos* better than *Count/SpatDist*
**3.** event presence in `Not` is the highest;
**4.** relation correctness in `And` is the highest;
**5.** parsimony score in `Not` is the highest;
**6.** event presence accuracy rate is below 1%;
**7.** relation correctness accuracy rate is below 1%;
**8.** An average of 2 redundant audio events;

Table 7: Key findings from experiments of TTA models on our RiTTA benchmark.

## 5.4 FINETUNING EXPERIMENTAL RESULT

We finetune Tango with the AdamW optimizer and follow the finetuning strategy outlined in Tango 2 (Majumder et al., 2024). The results, shown in Table 8, clearly demonstrate that finetuning Tango with relation aware datasets significantly improves its improves its ability to model

Table 8: Quantitative result across general and relation aware evaluation for Tango w/o finetuing.

| Model | General Evaluation | | | Relation Aware Evaluation (↑) | | | | mAMSR Across Four Main Relations | | | |
|---|---|---|---|---|---|---|---|---|---|---|---|
| | FAD↓ | KL↓ | FD↓ | mAPre | mARel | mAPar | mAMSR | Count | TempOrder | SpatDist | Compos |
| Tango (2023) | 10.79 | 90.26 | 39.46 | 11.13 | 2.27 | 9.88 | 3.10 | 0.16 | 3.44 | 0.82 | 8.10 |
| Tango (finetuing) | 4.60 | 23.92 | 27.03 | 21.23 | 10.78 | 20.35 | 48.67 | 8.04 | 324.10 | 1.88 | 44.42 |

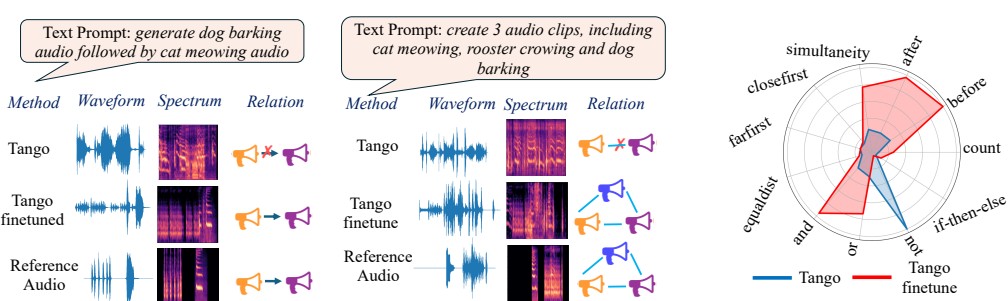

A. Qualitative audio events relation comparison on generated audio by Tango w/o finetuing

B. mAPre comparison for Tango w/o finetuing

Figure 7: Qualitative visualization comparison of Tango w/o finetuning (A) and mAPre w.r.t. 11 sub-relations. Listenable audios are provided in supplementary material.

audio event relations across both general and relation aware evaluations. This underscores the importance of benchmarking *RiTTA* with both comprehensive datasets and tailored evaluation metrics. Given that we finetuned only the latent diffusion model with a relatively small dataset (1.6 k pairs), further improvements can be expected by jointly finetuning other modules (*e.g.*, FLAN-T5) with a larger dataset. Moreover, the boosted performance indicates that audio events relation can indeed be modelled by TTA methods. We hope this benchmark and initial exploration will inspire more researchers to explore this area further.

Two qualitative examples are in Fig. 7 A. It is evident that the finetuned Tango successfully models the <before> relation (Table 1 and Fig. 1 show all existing TTA models fail on this case), and <count> relation. The mAPre score w.r.t. the 11 sub-relations is shown in Fig. 7 B (the mARel, mAPar, mAMSR are in Fig. I in Appendix). The results clearly indicates that finetuned Tango achieves significant improvements in target audio events presence across most relations, particularly in <Or>, <And>, <simultaneity>, <after> and <before>. The performance drop in <Not> relation may be attributed to the dataset preparation: as we pair <Not> relation with silent audio (all-zero waveforms), yet the text prompts might contain arbitrary audio events. Finetuning on such created data may confuse the model, leading to ambiguity in audio events generation. Further investigation is needed to address this challenge.

## 6 CONCLUSION AND FUTURE WORKS

Complex relationships within audio bring the world to life. While text-to-audio (TTA) generation models have made remarkable progress in generating high-fidelity audio with fine-grained context understanding, they often fall short in capturing the relational aspect of audio events in real-world. The world around us is composed of interconnected audio events, where audio event rarely occurs in isolation. Simply generating single sound sources is insufficient for producing realistic audio that reflects the richness of the world.

To analyze the capabilities of current state-of-the-art TTA generative models, we first conduct a systematic study of these models in audio event relation modeling. We introduce a benchmark for this task by creating a comprehensive relational corpus covering all potential relations in the real-world scenarios. Further, we propose new evaluation metric framework to assess audio event relation modeling from various perspectives. Additionally, we propose a finetuning strategy to boost existing models' ability in modelling audio events relation, and we show improvement across all relation metrics. Finally, we will release both the dataset and the code for the evaluation metrics, which will be useful for future research in this domain.

Going forward, our work provides a unique research opportunity to bring the world to life by exploring ways to generate long-term audio events to acoustically understand the physical world. Further, understanding the successes and failures of these models in generating such complex audio events is another promising research direction. This analysis could lead to further improvements in TTA models and their applications in areas such as virtual reality, cinema and immersive media.

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

Jeffrey M Zacks, Nicole K Speer, Khena M Swallow, Todd S Braver, and Jeremy R Reynolds. Event Perception: A Mind-Brain Perspective, 2007. 1

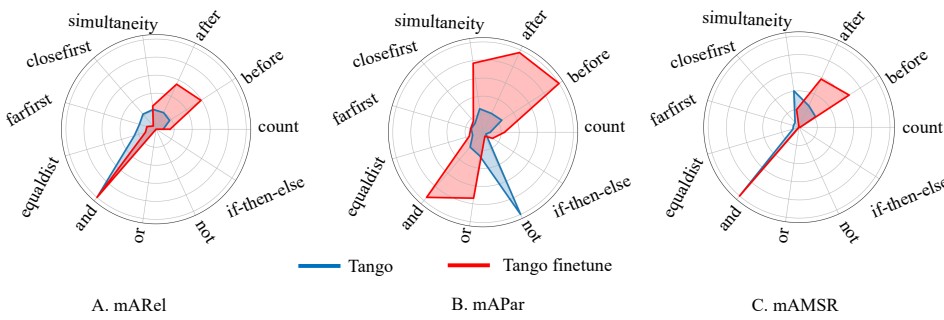

Figure I: The comparison of mARel, mAPar, mAMSR on Tango w/o finetuning.

| Methods | Setting |
|---|---|
| AudioLDM (S-Full) (2023a) | guidance_scale=5, random_seed=42, n_candidates=3 |
| AudioLDM (L-Full) (2023a) | guidance_scale=5, random_seed=42, n_candidates=3 |
| AudioLDM 2 (L-Full) (2023b) | guidance_scale=3.5, random_seed=45, n_candidates=3 |
| MakeAnAudio (2023b) | ddim_steps = 100, scale = 3.0 |
| AudioGen (2023) | model name: audiogen-medium |
| Tango (2023) | num_steps = 200, guidance=3, num_samples=1 |
| Tango 2 (2024) | num_steps = 200, guidance=3, num_samples=1 |

Table I: Detail setting for each TTA method

# A APPENDIX

## .1 FINETUNING PANNS AUDIO EVENT DETECTION MODEL ON OUR CURATED DATASET

## A AUDIO EVENT DETECTION MODEL FINE-TUNE

To detect the audio events from generated audio, we employ a pre-trained audio event detection model (in our case, we adopt PANNS (Kong et al., 2020b)) to detect all audio events, each detected event has class label with a confidence score, start time and end time. Analyzing these detected audio events can uncover various audio events relations (see Fig. 4 in the main paper).

The PANNS model (Kong et al., 2020b) is pre-trained on the large-scale 527 class AudioSet dataset (Gemmeke et al., 2017). It contains an audio tagging model and an audio event detection model. Directly applying the pre-trained detection model to detect audio events from our generated audios inevitably results in false positive and ambiguous detections. For instance, a *door slam* sound may be incorrectly detected as speech or music with high confidence scores. To mitigate the ambiguity and inaccuracies, we finetune the detection model ("Cnn14_DecisionLevelMax" variant) on our specially curated 100 k dataset by just tuning the last classification layer. Finally the finetuned model achieves mAP 0.57 on our curated 10k test sets, far outperforming the original model with mAP 0.43.

We based on the pretrained PANNS (Kong et al., 2020b) audio event detection model to finetune it on our curated 100 k audio training dataset. Each audio is 10 s long with sampling rate 16 kHz. Moreover, each audio randomly contains one to five audio events, each event has a random start time position in the 10 s long audio. The input is 10 s long audio waveform. The output is a confidence map of shape [20, 25], where 20 is the time steps with the temporal resolution 0.5 s and 25 is the audio event class number. Potential audio events are extracted from the confidence map by thresholding the confidence map, audio events with too short time duration (in our case, less than 0.5 s) are discarded. The training and testing datasets size are 100 k and 10 k respectively. We adopt Adam (Kingma & Ba, 2015) to train the model with initial learning rate 0.0001 but decays every 200 epochs with decaying rate 0.5. Finally, we train 350 epochs. The loss function is binary cross-entropy loss (BCE). On the testing dataset, the finetuned model achieves mAP 0.57. We use the finetuned audio event detection model to detection audio events from the generated audios.

### A.1 Existing TTA model Setting

We test 7 most recent TTA models: AudioLDM (Liu et al., 2023a) (two versions), AudioLDM 2 (Liu et al., 2024), MakeAnAudio (Huang et al., 2023b), AudioGen (Kreuk et al., 2023), Tango (Ghosal et al., 2023) and Tango 2 (Majumder et al., 2024). We depend on their released pre-trained model and use their recommended hyperparameter setting for benchmarking (from their Github page). The detailed setting for each TTA method is given in Table

### A.2 More Result on Tango Finetuning

The mARel, mAPar and mAMSR score w.r.t. 11 sub-relations is given in Fig. I.

