# OpenReview forum: "RiTTA: Modeling Event Relations in Text-to-Audio Generation"
_ICLR.cc/2025/Conference — Submitted to ICLR 2025_

### Official Review · Reviewer_Bn9E · 2024-10-27

**Soundness:** 3
**Presentation:** 3
**Contribution:** 3
**Rating:** 5
**Confidence:** 5

**Summary:**

This paper proposed an audio event relation corpus for text-to-audio generation and a new evaluation benchmark. TTA models finetuned in this corpus got better generation of the audio event relations.

**Strengths:**

The proposed corpus is important for research on audio event relations, which can be used for TTA with good quality and comprehension.

**Weaknesses:**

I have some concerns about this work:

1. The corpus only contains 25 categories, which seems to not easily generate to other categories with relations. Previous training metarials like AudioCaps and Clotho have over 100 classes.

2. The paper targets on mono-channel audio, but in this case, the spatial distance makes little sence. How can you accurately measure whether the event is from 1m or 7m? I think multi-channel audio would also be interesting, e.g. a car driving from left to right.

3. The FAD and FD used feature embeddings from pretrained on VGGish model. However, this model is known as its poor performance. Even though some baselines may use this tool, I think modern backbone models should be considered.

**Questions:**

1. Except the 25 categories, what are the results of other audio events with relations?  TTA is more like an open-set problem, so unseen labels' evaluation is needed.

2. What are the results on the previous test data from the baseline models? Will the finetuning affect the original results?

3. Can the model handle other lengths of audio instead of 10s?

---

> ### Author Response · Authors · 2024-11-21
> **Feedback by Authors 1**
>
> We appreciate the reviewer for spending time reviewing our work and further providing useful comments and advices for us to coninue improving our work.
>
> **Q1:** The corpus only contains 25 categories, which seems to not easily generate to other categories with relations. Previous training materials like AudioCaps and Clotho have over 100 classes.
>
> **A1:** We thank the reviewer for their careful examination of the corpus. Below, we address the concerns regarding the category number and its potential for generalization.
>
> 1. **Corpus Design and Intent**: The primary goal of RiTTA is not only try to cover a wide range of audio events but also guarantee that the introduced audio events are suitable for relation modeling. Therefore, the 25 categories are well-defined and meticulously constructed with a focus on ensuring high-quality, diverse and authentic (Sec. 3.3, L209-238) audio events that can effectively demonstrate relation modeling as detailed in section 3.1.
>
> Datasets like AudioCaps and Clotho with over 100 audio classes, serve broader purposes of general audio captioning and classification, they contain non-event audio (e.g., music audio, ambient environment sound) and audio without explicit eventness indication (e.g., ocean waves) that can not be used to model audio events relations. However, we agree that introducing more audio event categories will improve the benchmark robustness, we leave it as the next step endeavour.
>
> 2. **Adaptability and Expansion**: RiTTA framework is designed to scale, allowing for the inclusion of additional categories and relation types as more comprehensive data is curated. Moreover, we decouple events relation from event category, allowing us much flexibility to scale up the data from either relation or event category separately and independently. This adaptability ensures that the dataset can evolve to include more diverse categories akin to larger training materials like AudioCaps and Clotho.
>
> 3. **All Existing TTA Models Perform Poorly on Our Benchmark**. The detailed experiments show that all existing TTA models perform poorly on audio events relation we proposed in this paper, they failed to generate audio exhibiting even simple relations (Table 1, Figure 1). Therefore, our proposed benchmark is useful in terms of difficulty level and complex to benefit the TTA community to develop more advanced text-to-audio relation-aware models.
>
> 4. **Comparison with Prior Work**. The most relevant work CompoA [1], it simply discussed temporal relation and  contributed a benchmark with 400/200 test instances. our introduced benchmark is much more diverse in terms of relation classes and test instances (we have 8k test instances, and can be easily scaled up further).
>
> [1] Sreyan Ghosh et al., CompA: Addressing the Gap in Compositional Reasoning in Audio-Language Models. ICLR 2024.
>
>
> **Q2:** The paper targets on mono-channel audio, but in this case, the spatial distance makes little scene. How can you accurately measure whether the event is from 1m or 7m? I think multi-channel audio would also be interesting, e.g. a car driving from left to right.
>
> **A2:** We thank the reviewer for the in-depth examination of spatial distance indication from audio. It is true that mono-channel audio can not precisely identify the absolute position of audio events, which instead requires multi-channel audio as audio events' spatial position usually lies in inter-channel phase difference. We focus on mono-channel audio generation because 1) all existing text-to-audio (TTA) methods focus on mono-audio generation. 2) multi-channel audio generation require much more complex text description specifying each single audio event's spatial position, which remains as another future research topic and goes beyond the scope of this work.
>
> To make the spatial distance measurable, we propose to roughly measure the relative spatial distance between audio events, which is reflected in audio event's loudness that can be measured in mono-channel audio. This is also echoed by sub-relations we used in Spatial Distance relation: close-fist, far-fist, and equal distant (in Table 2) that can be measured by loudness in mono-channel audio. More details are given in section 3.1 L170-173, section 5.2 L415-427).
>
> In summary, precise spatial distance relation does require multi-channel audio. We reconcile this dilemma by approximating spatial distance by loudness variation, and we show the feasibility of our proposed spatial distance relation measurement in experiment Table 8.

---

> > ### Author Response · Authors · 2024-11-21
> > **Feedback by Authors 2**
> >
> > **Q3:** The FAD and FD used feature embeddings from pretrained on VGGish model. However, this model is known as its poor performance. Even though some baselines may use this tool, I think modern backbone models should be considered.
> >
> > **A3:** VGGish model [2] is a powerful CNN architecture trained on the large-scale AudioSet (currently the largest audio dataset), and has been unanimously adopted as audio feature extractor for all TTA models, including AudioLDM, AudioLDM 2, Tango, Tango 2. That is, we adopt VGGish model for feature embedding.
> >
> > [2] Shawn Hershey et al., CNN Architectures for Large-Scale Audio Classification. ICASSP 2017.

---

> > > ### Comment · Reviewer_Bn9E · 2024-11-25
> > >
> > > Thank you for addressing some of my concerns. However, I still cannot find any explanation or experiments in your method demonstrating how to generate new categories. Additionally, Tango and Tango 2 used a PANN model rather than a VGGish model to calculate the FAD and KL metrics, contrary to what you mentioned.
> > >
> > > Given these issues, I will maintain my current rating.

---

> ### Author Response · Authors · 2024-11-25
> **Further feedback to Reviewer Bn9E**
>
> Dear Reviewer,  thank you for your further comments. We would like to provide further feedback regarding your new comments.
>
> 1. **Comments**: Additionally, Tango and Tango 2 used a PANN model rather than a VGGish model to calculate the FAD and KL metrics, contrary to what you mentioned.
>
> **Feedback:** We double-checked Tango paper (link: https://arxiv.org/pdf/2304.13731). In page 6, it writes: in addition to FAD, we also used Frechet Distance (FD) as an objective metric. FD is similar to FAD, but it replaces the **VGGish classifier with PANN**. So for the FAD score it does use VGGish model. We followed the same setting to use VGGish model to calculte FAD and KL score.
>
> 2. **Comments**: I still cannot find any explanation or experiments in your method demonstrating how to generate new categories.
>
> **Feedback**: We are currently running experiments testing how the model performs when generating new categories, will report the result promptly. Thanks for your patience!
>
> Thanks you for spending time reading and commenting our feedback. We hope it discuss more with you if there is a need.

---

> > ### Author Response · Authors · 2024-11-29
> > **Further Feedback to Reviewer Bn9E**
> >
> > We thank the reviewer for the patience. Here we provide more feedback.
> >
> > 1. **Concern:** FAD/FD/KL score computation.
> >
> >    **Feedback:** We follow Tang, Tango 2 and other relevant TTA models to calculate FAD, FD and KL score. Specifically, we compute FD score based on PANNS extracted embedding, FAD and KL scored based on VGGish extracted embedding.
> >
> > 2. **Concern**: Experiment on generating new categories.
> >
> >     **Feedback:** We have introduced new audio categories for each of five audio categories. Specifically, for animal, we introduce frog croaking, lion roaring. For human, we introduce clapping and finger snapping. For human-object interaction, we introduce bag-zipping and beer-cap-opening. For machineary, we introduce hair dryer and vacuum cleaner. Then we use RiTTA text and relation generation strategy to generate the 300 text prompts and corresponding reference audio. By running inference on our fine-tuned Tango model, we report the FAD/FD/KL score comparison in the following table.
> >
> >    | Category Info   | FAD ($\downarrow$)   | KL ($\downarrow$)  | FD ($\downarrow$)   |
> >    |------------|------------|------------|------------|
> >    | Existing 25 categories | 4.60 | 23.92 | 27.03 |
> >    | New 10 categories | 11.43 | 173.00 | 94.59 |
> >
> >     We can observe from the table that asking the model to generate previously-never-seen categories inevitably leads to performance reduction, and it reaches the performance on par with AudioSet/AudioCap pretrained generative model's performance on our created dataset (see Table 5 in the main paper). We also listened to 50 generated new categories audios, and found most of the generated audios just contain one audio event and they listened very unnatural. Therefore, it remains as a future challenge to design novel TTA framework to automatically incorporate new audio event categories.  We think it still remains a challenge for all TTA models to learn to generalize to new audio event categories. Existing AudioSet and AudioCaps training set and testing set have audio events category overlap.
> >
> > 3. **Concern**: Fine-tuned model performance on previous AudioCaps testing set.
> >
> >    We report the fine-tuned Tango's performance on AudioCaps testing set in the following table (the Initial Tango (Trained on AudioCaps) result is from Tango paper Table 1, see https://arxiv.org/pdf/2304.13731):
> >
> >    | Model   | FAD ($\downarrow$)   | KL ($\downarrow$)  | FD ($\downarrow$)   |
> >    |------------|------------|------------|------------|
> >    | Initial Tango (Trained on AudioCaps) | 1.59 | 1.37 | 24.52 |
> >    | Fine-tuned Tango (finetuned on our RelationAware Data) | 13.39 | 169.85 | 37.39 |
> >
> >    From this table, we can conclude that finetuning Tango model on our introduced relation-aware dataset will reduce the model performance on the previous task before finetuning. We think this is understandable because finetuning on new task and new datasets in a data-driven way will surely adjust the model to handle new task and gradually loose its capability in original tasks.

---

### Official Review · Reviewer_yPb8 · 2024-10-31

**Soundness:** 3
**Presentation:** 3
**Contribution:** 2
**Rating:** 5
**Confidence:** 1

**Summary:**

1.	Benchmark Construction: Created an audio event relation corpus for TTA tasks, covering common spatial, temporal, count, and compositional relations. Additionally, an audio event category corpus encompassing various everyday sounds was established to provide foundational data for studying event relation modeling in TTA.
2.	Evaluation Framework: Proposed a multi-stage relation-aware evaluation framework (MSR-RiTTA) that quantifies a model’s ability to capture audio event relations from multiple perspectives, offering more targeted metrics than existing TTA evaluation standards.
3.	Fine-Tuning Framework: Developed a fine-tuning strategy aimed at enhancing current TTA models’ capacity to model audio event relations, showing significant improvement in benchmark tests.

**Strengths:**

The paper presents a substantial contribution to the field of Text-to-Audio (TTA) generation by introducing the RiTTA framework, which systematically addresses the modeling of audio event relations—a previously underexplored area. It proposes an innovative multi-stage, relation-aware evaluation framework (MSR-RiTTA) to more accurately assess the performance of TTA models in capturing complex spatial, temporal, count, and compositional relationships. The research is rigorous in its dataset construction and experimental validation, demonstrating clear improvements in TTA models’ relation modeling abilities. This work stands out for its originality, methodological clarity, and significant potential impact, expanding TTA applications in areas like virtual reality and immersive media.

**Weaknesses:**

The paper benchmarks seven recent TTA models (such as AudioLDM and Tango) but focuses primarily on single-strategy models, and without testing in more complex scenarios like audio events in virtual reality or interactive environments. This limits the framework’s applicability across different model types and environments. A wider variety of model types, such as multimodal generation models or models optimized for extended audio sequences, and more diverse application datasets, like multi-event VR audio environments are recommended.

The description of the fine-tuning strategy is too general, lacking detailed information on training settings, parameter tuning strategies, and specific experimental procedures. The lack of details such as specific hyperparameter settings, number of iterations, and choice of loss function also weakens persuasiveness.

**Questions:**

Refer to the weaknesses.

---

> ### Author Response · Authors · 2024-11-21
> **Feedback by Authors**
>
> We sincerely thank the reviewer for the useful comments and suggestion for future expansion.
>
>
> **Q1:** The paper benchmarks seven recent TTA models (such as AudioLDM and Tango) but focuses primarily on single-strategy models, and without testing in more complex scenarios like audio events in virtual reality or interactive environments. This limits the framework’s applicability across different model types and environments. A wider variety of model types, such as multimodal generation models or models optimized for extended audio sequences, and more diverse application datasets, like multi-event VR audio environments are recommended.
>
> **A1:** We appreciate the reviewer’s insightful suggestion.
>
> 1. **Focus on Single-Strategy TTA Models**: We agree that our current benchmarking focuses on recent TTA models (e.g., AudioLDM, Tango, Tango 2) that utilize a single-strategy approach. Our goal in this initial study was to establish a robust foundation by systematically evaluating models designed specifically for text-to-audio generation. By benchmarking these models with consistent metrics, we aimed to create a controlled environment for meaningful comparisons (Sec. 5.3). It is worth noting that our focus in this work is to benchmark audio events relation modeling in TTA task, and we found all existing TTA models perform poorly on all proposed relations. So our introduced benchmark makes significant contribution to benefit TTA community to develop more advanced audio events relation aware frameworks.
>
> 2. **Complex Scenarios and Extended Environments**: We acknowledge that testing in more complex scenarios such as virtual reality (VR) or interactive environments could provide valuable insights into a framework's robustness. To accommodate multi-event VR audio or extended interaction-based tests, we can base on current relation and audio events corpus to involve more complex AR/VR context information for text description generation, and further design more complex techniques to generate the AR/VR aware audio.  For example,
>
> * Generate a dog barking audio, followed by cat meowing audio in a 3D spacious and empty room with pedestrian walking around.
>
> * Create a vegetable chopping audio and toilet flushing audio simultaneously and each audio resembles well with the indoor environment's kitchen and restroom positions.
>
> In conclusion, we appreciate the reviewer’s thoughtful suggestions and acknowledge that expanding our benchmarking to more complex, multimodal, and interactive environments will strengthen the framework. We view this as an ongoing project that will evolve to address these richer applications and extend the study’s reach to more diverse model types and use cases.
>
> **Q2:** The description of the fine-tuning strategy is too general, lacking detailed information on training settings, parameter tuning strategies, and specific experimental procedures. The lack of details such as specific hyperparameter settings, number of iterations, and choice of loss function also weakens persuasiveness.
>
> **A2:** We thank the reviewer for point out the lack of details in fine-tuning strategy. We use Adam optimizer with a learning rate of $3\times10^{-5}$, batch size of $16$, SNR gamma value of $5$. The training was performed for $40$ epochs on $4$ A$100$ GPUs. The rest of the hyperparameters were kept the same as base model. We will include these specific details in the revised version of the paper.

---

> > ### Author Response · Authors · 2024-11-29
> > **Thanks for Reviewing our paper**
> >
> > Dear Reviewer yPb8,
> >
> > We sincerely appreciate your efforts in providing constructive reviews. During the rebuttal period, we have tried our best to provide feedback regarding your concern. If you have any further concern, welcome to set up further discuss. We will be more than happy to provide more feedback on them.
> >
> > Thank you again for your constructive comment.
> >
> > Best,
> >
> > Authors

---

### Official Review · Reviewer_rsts · 2024-11-01

**Soundness:** 2
**Presentation:** 3
**Contribution:** 2
**Rating:** 5
**Confidence:** 5

**Summary:**

This paper primarily introduces a benchmark for the relationships between multiple audio events, with its main contribution being the creation of a new dataset and the establishment of several novel evaluation metrics.

**Strengths:**

1. Compared to previous work, the authors' most significant contribution lies in the development of a comprehensive dataset that assesses the relationships between multiple audio events.
2. Using temporal relationships as an example, the authors provide a detailed explanation of the dataset construction process and propose new evaluation metrics. They also conduct experiments using both the pre-trained TTA model and the fine-tuned version.

**Weaknesses:**

1. I am unclear as to why RiTTA was submitted to the generative model track, as the primary contribution clearly stems from the new dataset and evaluation metrics related to the benchmark, rather than from advances in generative modeling.
2. In terms of innovation, the authors are not the first to explore audio event relationship modeling. RiTTA extends temporal relationships to four types of relationships, but the overall process appears to be data-driven. However, merely using the TTA model to model complex relationships is insufficient. As demonstrated in Figure 7, there is a performance decline with "not" relationships. I was expecting a novel TTA framework capable of jointly modeling relationships between different audio events. Unfortunately, RiTTA only fine-tunes the TTA model, which limits its novelty.
3. Regarding the core contribution—the dataset—there are several potential shortcomings. First, the dataset construction process does not seem to require significant effort. Second, the dataset itself feels somewhat toy. While the authors attempt to enhance its randomness, expanding each example to five events via ChatGPT is far from sufficient (only 5?); 500 events might be a more appropriate target. Additionally, the use of only one-channel audio for spatial relationships could limit the dataset’s diversity.

**Questions:**

1. The authors provide a detailed explanation of the dataset construction process using temporal relationships as an example, but I am still curious about how other relationships, such as spatial relationships, were constructed. How were the corresponding text-audio pairs created for these? Further clarification would be beneficial.
2. Should out-of-domain scenarios be considered when forming the test set?
3. I could not find the RiTTA dataset in the supplemental materials.

---

> ### Author Response · Authors · 2024-11-21
> **Feedback by Authors 1**
>
> We thank the reviewer for the constructive feedback and suggestions, which are really helpful for us to continue to improve our work. Regarding your concerns, we provide one-by-one feedback.
>
> **Q1:** I am unclear as to why RiTTA was submitted to the generative model track, as the primary contribution clearly stems from the new dataset and evaluation metrics related to the benchmark, rather than from advances in generative modeling?
>
> **A1:** Thank you for highlighting your concern regarding the appropriate track for our submission. We agree that RiTTA has introduced significant contributions audio events relation modeling in TTA task, including a new benchmark, novel evaluation metric and a fine-tuning strategy. These aspects indeed align well with the scope of new datasets and evaluation metrics.
>
> However, we would like to emphasize that we have conducted in-depth and systematic research audio events relation problem that shows potential not only merely in TTA task, but also in a much broad spectrum involving audio events relation analysis (e.g., holistic acoustic scene understanding, reasoning). We submit it to generative model because existing TTA models perform poorly in audio events relation modeling (Table 1 and Fig. 1).
>
> Moreover, we find some other ICLR accepted papers that also contributed benchmark and corresponding neural network framework, such as [1] and [2], We hope this clarifies our rationale for submitting to the generative model track and underscores the broader impact of our contributions on the field of text-to-audio generation.
>
> [1] Sreyan Ghosh et al., CompA: Addressing the Gap in Compositional Reasoning in Audio-Language Models. ICLR 2024.
>
> [2] Garrett Tanzer et al., A Benchmark for Learning to Translate a New Language from One Grammar Book. ICLR 2024.
>
> **Q2:** In terms of innovation, the authors are not the first to explore audio event relationship modeling. RiTTA extends temporal relationships to four types of relationships, but the overall process appears to be data-driven. However, merely using the TTA model to model complex relationships is insufficient. As demonstrated in Figure 7, there is a performance decline with "not" relationships. I was expecting a novel TTA framework capable of jointly modeling relationships between different audio events. Unfortunately, RiTTA only fine-tunes the TTA model, which limits its novelty.
>
> **A2:** We acknowledge the concerns raised and would like to address them with clarifications based on our methodology and results:
>
> 1.  **Relation to prior work**: Although prior works have touched upon one relation: temporal relation, RiTTA provides a systematic study that sets it apart by establishing a comprehensive benchmark specifically for modeling audio event relations. We scale up the temporal relation to 4 different relations, accompanied by new audio events, new evaluation metric, new benchmark. That is, RiTTA isn't an incremental work based on prior temporal relation work.
>
> 2. **Fine-Tuning Strategy**: The critique regarding the novelty of fine-tuning is understood. However, we emphasize that the RiTTA framework was not designed to merely apply existing models but to adapt and expand upon them with targeted fine-tuning. Our method fine-tunes the latent diffusion model component of Tango, enabling it to better model audio events relations (Section 4). This nuanced strategy yielded significant improvements across key evaluation metrics, showcasing our approach’s practical impact (Table 8).
>
> 3. **On "Not" Relation**: The reviewer’s observation regarding "not" relationships is valid. We conducted further analyses that indicated this outcome was tied to the silent audio data representation paired with <Not> prompts, leading to potential model ambiguities (Section 5.4). Moreover, **it is worth noting that not being able to model "Not" relation is an intrinsic challenge for all TTA models, because the text just tells what to not generate, no guidance what to generate at all.** Future work could explore how to handle this situation.
>
> We hope these points illustrate the innovative aspects and address concerns regarding RiTTA’s contributions to TTA models. We are committed to further refining the framework to enhance its robustness and clarity.
>
>
> **Q3:** Dataset shortcomings. First, the dataset construction process does not seem to require significant effort. Second, the dataset itself feels somewhat toy. While the authors attempt to enhance its randomness, expanding each example to five events via ChatGPT is far from sufficient (only 5?); 500 events might be a more appropriate target. Additionally, the use of only one-channel audio for spatial relationships could limit the dataset’s diversity.
>
> **A3:** We sincerely thank the reviewer for their detailed feedback and constructive points regarding the dataset.  (continue in Feedback 2).

---

> > ### Author Response · Authors · 2024-11-21
> > **Feedback by Authors 2**
> >
> > **A3:** We sincerely thank the reviewer for their detailed feedback and constructive points regarding the dataset.
> >
> > 1. **Effort in Dataset Construction**: The dataset development process for RiTTA involved several methodical and resource-intensive steps: First, **Corpus Creation and Data Curation**: We constructed an audio event relation corpus by identifying and categorizing real-world audio event relationships across Temporal Order, Spatial Distance, Count, and Compositionality. This was not a trivial task, as it required a deep understanding of audio event interaction and context, as outlined in Sections 3.1 and 3.2. Second, **Diverse Seed Audio Collection**: We sourced and curated audio samples from various platforms (e.g., Freesound.org, VCTK dataset) and ensured diversity by selecting multiple audio variations for each event type. This enriched the dataset’s representativeness and complexity.
> >
> > 2. **"Toy" Data Concern**: Despite the reviewer's perception of the benchmark as ``toy'', it serves as a foundational benchmark meticulously designed for audio events relation modeling in TTA task. First, comparing with the most relevant work CompoA [1] which discussed temporal relation with just 400 and 200 test instances, our introduced benchmark is several magnitudes larger than it. Second, we show in the paper that all existing TTA models fail on the benchmark we have presented. So the benchmark is already beneficial enough for TTA research community to push forward the audio events relation in TTA task. Third, **Adaptivity and Scalability**. As we decouple audio events relation from audio events category, the benchmark construction process can be easily scaled up to incorporate extra relation and audio events so as to augment with much more complexity. Therefore, the benchmark we have introduced isn't fixed, but instead can be easily scaled up for more complex audio events relation modeling as the relevant technology has developed.
> >
> > 3. **Mono-Channel Concern**: We focus on mono-channel in this paper because all existing TTA models generate mono-channel audio and mono-channel audio is sufficient enough to model the four relations. Moreover, we approximate spatial distance relation by measuring the loudness (section 3.1 L170-173, section 5.2 L415-427). Multi-channel audio generation requires extra text description and novel generative model, it goes beyond the scope of this work.
> >
> > **Q4:** Should out-of-domain scenarios be considered when forming the test set?
> >
> > **A4**: Our main contribution is the benchmarking and evaluation on existing TTA models with zero-shot setting. We are running another out-of-domain experiment, will report the result once it comes out.
> >
> > **Q5:** I could not find the RiTTA dataset in the supplemental materials.
> >
> > **A5**: Thanks for pointing this out. We have provided more detailed RiTTA dataset description in the refined supplementary material.

---

> > > ### Comment · Reviewer_rsts · 2024-11-26
> > >
> > > Thank you for the author's response. However, it is evident that the core issues raised in my weaknesses 2 and 3 have not been adequately addressed. The dataset, compared to previous work, only involves a simplistic extension of relationships and is highly limited both in terms of size and scope, making it more of a toy example. Furthermore, no new model design has been introduced. As a result, I will maintain my score and increase my confidence level.

---

### Author Response · Authors · 2024-11-29
**Feedback to All Reviewers**

We appreciate the reviewers' constructive comments and feedback.  We have provided feedback to each reviewer separately. Here we further provide a general feedback to all reviewers.

1. Benchmark scale and comparision with previous discussion on audio events relation.

   **Feedback:** **First**, comparing with prior work (like CompoA) that just discussed temporal relation and 400 and 200 data size, our introduced benchmark incorporated four more main relations, 20 more subrelations, and the corresponding data size is several magnitudes larger (currently, 8,000, and can further be scaled up). **Second**, we extensively show in the paper that all existing TTA models fail on the benchmark we have presented. So the benchmark in its current format is already beneficial enough for TTA research community to investigate deeper for the audio events relation modeling in TTA task. **Third**, Adaptivity and Scalability. We decoupled audio event relation from audio events category, the benchmark construction process can be easily scaled up to incorporate extra relation and audio events.

2. RiTTA dataset is missing in supplementary (as per to Reviewer rsts).

   In the updated supplementary material, we have provided RiTTA **1. seed audios,  2. created events relation aware reference audios, 3. GPT-augmented text prompts.**. With the updated material, we hope the reviewer can have a direct understanding of our benchmark.

3. More experiment of our fine-tuned model (as per to Reviewer Bn9E).

   Following the requirements, we have run two extra experiments on the fine-tuned model, which provide reviewers with in-depth understanding of our work.

---

### Meta-Review · Area_Chair_cZT8 · 2024-12-20

**Metareview:**

The paper's main contribution is a new audio dataset and benchmark, which can be used for "text to audio" (TTA) experiments and evaluation. In conjunction with the benchmark, the paper discusses dataset construction, and some analysis that shows that existing text to audio models fail on the new benchmark. Challenging issues (which the reviewers would like to see), such as generation of unseen audio events, more complex event relationship patterns, etc are left for future work.

The main contributions are:
- Created an audio event relation corpus for TTA tasks, covering common spatial, temporal, count, and compositional relations. Additionally, an audio event category corpus encompassing various everyday sounds was established to provide foundational data for studying event relation modeling in TTA.
- Proposed a multi-stage relation-aware evaluation framework (MSR-RiTTA) that quantifies a model’s ability to capture audio event relations from multiple perspectives, offering more targeted metrics than existing TTA evaluation standards.
- Developed a fine-tuning strategy aimed at enhancing current TTA models’ capacity to model audio event relations, showing significant improvement in benchmark tests.

Strengths
- The paper treats an interesting problem and any progress towards complex audio generation (and this includes datasets, evaluation method, and running benchmarking) is a worthwhile contribution
- The paper is generally well written and easy to understand, the paper is accessible

Weaknesses
- it is not clear if the paper should not be in the dataset/ evaluation track, since its technical contributions are slim (even if not entirely absent)
- Audio Event Relationship modeling is not entirely new, authors are not the first to propose that. RiTTA presents incremental improvements. The paper would be much stronger if the dataset had bigger complexity, bigger size, and/ or a better model that can solve the described challenges

Summary
While reviewers are sympathetic to the papers basic premise, they agree that the presented contributions do not quite make the bar for publication at ICLR, for both (generative) modeling as well as dataset/ evaluation track.

**Additional Comments On Reviewer Discussion:**

Reviewers discussed with authors. Authors stressed the contributions of their work, but reviewers stated they would need to see substantial expansions of the work (like discussed above), before they would consider raising their scores.

---

### Decision · Program_Chairs · 2025-01-22

Reject